# CAIX-Mediated Control of LIN28/*let-7* Axis Contributes to Metabolic Adaptation of Breast Cancer Cells to Hypoxia

**DOI:** 10.3390/ijms21124299

**Published:** 2020-06-16

**Authors:** Adriana Gibadulinova, Petra Bullova, Hynek Strnad, Kamil Pohlodek, Dana Jurkovicova, Martina Takacova, Silvia Pastorekova, Eliska Svastova

**Affiliations:** 1Department of Tumor Biology, Institute of Virology, Biomedical Research Center, Slovak Academy of Sciences, Dubravska cesta 9, 845 05 Bratislava, Slovakia; virugiba@savba.sk (A.G.); petra.bullova@ki.se (P.B.); martina.takacova@savba.sk (M.T.); virusipa@savba.sk (S.P.); 2Institute of Molecular Genetics, The Czech Academy of Sciences, Vídeňská 1083, 142 20 Prague, Czech Republic; strnad@img.cas.cz; 32nd Department of Gynecology and Obstetrics, Faculty of Medicine, Comenius University of Bratislava, Ruzinovska 6, 821 01 Bratislava, Slovakia; kpohlodek@gmail.com; 4Cancer Research Institute, Biomedical Research Center, Slovak Academy of Sciences, Dubravska cesta 9, 845 05 Bratislava, Slovakia; Dana.Jurkovicova@savba.sk

**Keywords:** carbonic anhydrase IX, hypoxia, LIN28/*let-7* axis, metabolism

## Abstract

Solid tumors, including breast cancer, are characterized by the hypoxic microenvironment, extracellular acidosis, and chemoresistance. Hypoxia marker, carbonic anhydrase IX (CAIX), is a pH regulator providing a selective survival advantage to cancer cells through intracellular neutralization while facilitating tumor invasion by extracellular acidification. The expression of CAIX in breast cancer patients is associated with poor prognosis and metastases. Importantly, CAIX-positive hypoxic tumor regions are enriched in cancer stem cells (CSCs). Here we investigated the biological effects of *CA9*-silencing in breast cancer cell lines. We found that CAIX-downregulation in hypoxia led to increased levels of *let-7* (lethal-7) family members. Simultaneously with the increase of *let-7* miRNAs in CAIX-suppressed cells, LIN28 protein levels decreased, along with downstream metabolic pathways: pyruvate dehydrogenase kinase 1 (PDK1) and phosphorylation of its substrate, pyruvate dehydrogenase (PDH) at Ser-232, causing attenuation of glycolysis. In addition to perturbed glycolysis, CAIX-knockouts, in correlation with decreased LIN28 (as CSC reprogramming factor), also exhibit reduction of the further CSC-associated markers NANOG (Homeobox protein NANOG) and ALDH1 (Aldehyde dehydrogenase isoform 1). Oppositely, overexpression of CAIX leads to the enhancement of LIN28, ALDH1, and NANOG. In conclusion, CAIX-driven regulation of the LIN28/*let-7* axis augments glycolytic metabolism and enhances stem cell markers expression during CAIX-mediated adaptation to hypoxia and acidosis in carcinogenesis.

## 1. Introduction

Carbonic anhydrase IX (CAIX) is a hypoxia-induced transmembrane protein that catalyzes the reversible hydration of carbon dioxide into bicarbonate ions and protons [1]. The reaction significantly contributes to the neutralization of intracellular pH, as bicarbonate ions are directly transported into the cell. CAIX therefore plays a crucial role in the maintenance of favorable intracellular pH (pHi) and provides a selective advantage for cancer cells, which are better adapted for survival in such conditions, while it also promotes cancer progression [2,3]. One of the consequences of hypoxia is the up-regulation of glycolysis and the associated production of lactic acid. Gatenby et al. [4] provided evidence that adaptation to hypoxia and acidosis are necessary steps in the later phase of the carcinogenesis and may represent key events of somatic evolution of breast tumors in the transition from in situ to invasive cancer. We hypothesize that CAIX as a pH regulator in hypoxic cancer cells, could participate in the control of metabolic pathways, especially since several glycolytic enzymes are pH-sensitive, and alkaline pH has been reported to promote glycolysis [5].

Recently, studies have identified a connection between the hypoxic feature of the neoplastic microenvironment and a specific group of microRNAs [6]. An interesting case is the members of the *let-7* family, which seem to exhibit the opposite response during hypoxia, considering different findings with regard to different cell types and different laboratories. Hebert et al. identified *let-7g*, 7*e*, and -7*i* as hypoxia-upregulated in 1% O_2_ [7], whereas *let-7a*, *c*, *d*, *e*, *f*, and *g* levels have been reported to decrease during hypoxia inducement by desferrioxamine [8]. Since all *let-7* family members are regulated by LIN28 through blocking of their processing into mature miRNAs [9] and LIN28 itself is downregulated by *let-7*, existing regulatory *let-7*/LIN28 feedback loop exhibits similar contrasting patterns of LIN28 expression during hypoxia. LIN28 (a homologue of the *Caenorhabditis elegans lin-28* gene) is an evolutionarily conserved RNA-binding protein with a critical role in cellular development [10] and in control of embryonic stem cell pluripotency and with a recently determined predominant function of an oncogene [11,12]. Mammals have two LIN28 homologs; LIN28A (called LIN28) and LIN28B (with an additional nuclear localization signal, and therefore primarily located in the nucleus). The expression of LIN28A/B is upregulated in different malignancies, however, the studies seldom compare the expression and functions of both homologs [13].

Aberrant expression of LIN28 and *let-7* facilitates aerobic glycolysis, or the Warburg effect, in cancer cells [14,15,16,17]. The Warburg effect, metabolic adaptation characteristic for the majority of human cancers, utilizes glucose and facilitates the uptake and incorporation of nutrients into the biomass [18]. This metabolic switch requires attenuation of oxidative phosphorylation with concomitant enhancement of glycolytic metabolism. Pyruvate dehydrogenase kinase 1 (PDK1) inhibits the activity of pyruvate dehydrogenase (PDHA) and converts pyruvate to acetyl-CoA, allowing the use of pyruvate pool in glycolysis [19,20]. LIN28 enhances, whereas *let-7* suppresses, aerobic glycolysis by targeting pyruvate dehydrogenase kinase 1 (PDK1), independently of hypoxia- or hypoxia-inducible factor-1 (HIF-1) [14]. Moreover, PDK1 was revealed as a key executor of LIN28-driven proliferation of cancer cells through direct potentiation of cellular metabolism [15,21]. Increasing evidence suggests that LIN28 may also be a master regulator controlling the pluripotency of embryonic stem cells and cancer cells [22].

Here we investigated the biological effects of gene silencing of CAIX in breast cancer cell lines. To avoid cell-line-specific effects of CAIX suppression, we used three different breast cancer cell lines. All used cell lines express low or no amount of CAIX protein in normoxia, but its expression is strongly upregulated in hypoxia (with moderate upregulation by high cell density). We show that suppression of CAIX affects the *let-7*/LIN28 axis with the effect on associated metabolic pathways and stem cell reprogramming.

## 2. Results

### 2.1. Suppression of CAIX Affects the Let-7/LIN28 Axis

To elucidate an impact of CAIX silencing on microRNAs and protein-coding genes’ expression profile in the context of hypoxia and cancer progression, we suppressed CAIX by transient siRNA-mediated knockdown in breast cancer MCF7 cell line (with CAIX expression upregulated by hypoxia). Gene expression differences between hypoxic MCF7 cells transiently transfected with control-siRNA and CA9-siRNA were analyzed by Microarray (Human Gene 1.0ST Array, Affymetrix). Most notably, siRNA-mediated suppression of CAIX showed (Table 1) increased levels of the *let-7* family microRNA members namely *let-7*d (logFC 0.51), *let-7*c (logFC 0.5) and *let-7*f-1 (logFC 0.43) and *NF-κB* (logFC −0.83). These results were validated by PCR (Figure 1D,E).

Interestingly, oncogenic regulation of *let-7* microRNAs has been demonstrated in several human malignancies. Expression of the *let-7* family members is significantly reduced in human cancers, including breast cancer. Since *let-7d*, -*c*, and –*f* decrease during hypoxia [6], and a double-negative feedback loop exists between LIN28 and *let-7* [9,12], we hypothesized that LIN28 could be induced by hypoxia and looked at LIN28 expression in hypoxic control and CAIX–depleted cells. We found that hypoxia increased LIN28 protein levels in all three tested breast cancer cell lines with the most profound elevation in MCF7 cells (Figure 1). The transient silencing of *CA9* in hypoxic MCF7 cells led to a decrease in LIN28 protein and mRNA levels (Figure 1A,B). MDA-MB-231 display drop of LIN28 mRNAs in si*CA9* hypoxic cells, as well (Figure 1B).

To confirm that the revealed changes in LIN28 expression were specifically caused by CAIX downregulation in siCA9-cells, we used another approach to abrogate CAIX expression. We prepared CAIX knockout cells (MDA-MB-231 and T47D) using the CRISPR/Cas9 system and observed a significant decrease of LIN28 protein level in MDA-MB-231-CA9-KO and T47D-CA9-KO cells cultured in hypoxia (Figure 1C). Regarding literature data, we confirmed that LIN28B homolog is expressed in MCF7 and MDA-MB-231, whilst LIN28A homolog is highly expressed in T47D. Therefore, we detected the particular LIN28A or B homologs with different antibodies for different cell lines. The particular LIN28A/B homologs were detected with different antibodies for individual cell lines. MDA-MB-231 and T47D CAIX-knockout cells displayed a reduction in LIN28B and LIN28A proteins, respectively. Importantly, the impact of CAIX downregulation on *let-7* family members was confirmed in CAIX-knockout cells (Figure 1D).

### 2.2. CAIX Suppression Reduces Glycolysis by Targeting PDK1

One of the consequences of hypoxia is upregulation of glycolysis and the associated production of lactic acid. Several glycolytic enzymes are pH-sensitive, and alkaline pH has been reported to promote glycolysis [5]. Therefore, we hypothesize that CAIX as a pH regulator in hypoxic cancer cells could participate in the control of metabolic pathways. In assessing the relationship between CAIX and LIN28 in cellular metabolism, we examined several key enzymes involved in cancer metabolism using Western blotting and real-time PCR. In MCF7 cells where PDK1 is nearly undetectable in normoxia, but strongly induced by hypoxia, CA9-silencing (siCA9) led to a significant reduction in PDK1 protein, as well as in mRNA levels (Figure 2A,C,D). PDK1 inhibits pyruvate dehydrogenase (PDH) by phosphorylation of its E1a subunit and thus promotes glycolysis by decreasing mitochondrial function. Accordingly, significantly decreased phosphorylation of PDH at Ser-232 by PDK1 was also observed in siCA9 cells (Figure 2B).

Consistently with our siCA9 results, CAIX knockout cell lines MCF7-CA9-KO, MDA-MB-231-CA9-KO, and T47D-CA9-KO cultured in hypoxia confirmed the decreased levels of LIN28A/B and PDK1 proteins. Moreover, transient overexpression of CAIX in CA9-KO cells using pcDNA3.1-FL-CA9 plasmid, led to a significant reversion of PDK1 and LIN28 expression in hypoxia (Figure 2E). Lactate dehydrogenase A (LDHA) expression was not significantly changed in any of these experiments. Immunofluorescence of hypoxic CAIX-knockout MDA-MB-231 cells also showed reduced LIN28 and slightly increased LIN28 in FL-CA9-transfected KO cells (Figure 2F).

To assess metabolic consequences of reduced Ser-232-PDH phosphorylation in hypoxia, we measured lactate production in culture media of CAIX-suppressed cells and subsequently in CA9-KO clones with restored CAIX expression twice by an enzymatic assay (Lactate Assay Kit, Sigma), and moreover, once by Lactate-GloTM Assay (Promega). Extracellular lactate levels decreased to 78.8% in MDA-MB-231-CA9-KO, 83.5% in T47D-CA9-KO, and 90% in MCF7-CA9-KO cells compared to their hypoxic control counterparts. Moreover, transient overexpression of CAIX restored lactate production in all tested transfected CA9-knockouts (Figure 2G).

### 2.3. CAIX Depletion Reduces NF-κB Expression and Transactivation

One of the key transactivators of LIN28 in transformed cancer cells is NF-κB. NF-κB directly activates LIN28 expression with consecutive downregulation of *let-7* levels [11]. According to our microarray results, CAIX silencing decreased *NF-κB* (logFC −0.89) (Table 1) and consistently with that, LIN28 protein level was decreased (Figure 1A–D), while *let-7* family members were upregulated (Table 1).

To confirm the effects of CAIX downregulation on NF-κB, we used a NF-κB luciferase reporter assay for monitoring the activity of the NF-κB signaling pathway in hypoxic cells expressing the CAIX protein in comparison to CAIX-knockout cell lines MCF7-CA9-KO and MDA-MB-321-CA9-KO. Luciferase assay showed significantly reduced NF-κB transactivation in CA9-KO MCF7 to 84.7%, as well as MDA-MB-231 cells to 73%, compared to their control counterparts (Figure 3A). In the case of hypoxic MCF7-siCA9 cells, NF-κB activity decreased to 57%. Our results are in agreement with the fact that CAIX expression is required for the activation of NF-κB in hypoxic breast cancer cells [23].

### 2.4. Extracellular Acidosis and Inhibition of CAIX Enzymatic Activity Affects NF-κB, LIN28B and PDK1 Expression

It is well known that extracellular acidosis and hypoxia can activate NF-κB, which promotes cell invasion [24,25,26,27]. CAIX as an established pH regulator with the role in extracellular acidification and intracellular neutralization in hypoxic cancer cells is vitally connected with hypoxic/acidic tumor microenvironment. Thus, there is a strong rationale for the role of CAIX in the regulation of NF-κB. We showed that depletion of CAIX leads to decreased NF-κB expression, as well as activity in MCF7, and MDA-MB-231 hypoxic cells (Table 1, Figure 3A). Thus, we also tested the effect of extracellular acidosis on NF-κB expression. Incubation of cells in acidic medium (pH 6.7) for 48 h led to significantly increased NF-κB expression in MCF7, as well as MDA-MB-231 cells (Figure 3B). Moreover, extracellular acidosis increased the expression of CA9, LIN28, and PDK1 in both tested cell lines (Figure 3C). Similarly, we evaluated the impact of CAIX enzymatic inhibition by homosulfanilamide (HSFA) on the set of the same genes and showed that NF-κB, LIN28, and PDK1 were significantly downregulated, as detected by qPCR (Figure 3B). We assume that consequences of CAIX induction in hypoxia, which is decreased (acidic) extracellular pH, and simultaneously increased (buffered) intracellular pH, enhance NF-κB expression/activity.

### 2.5. CAIX Knockout Decreases Cell Proliferation

Overexpression of LIN28 has been shown to promote cancer cell proliferation, and PDK1 is critical for LIN28A/B-mediated cancer proliferation as well [12]. Thus, we tested the effect of CAIX elimination on cell proliferation of MDA-MB-231-CA9-KO, T47D-CA9-KO in a real-time setting using the xCELLigence system. We compared MDA-MB-231 and T47D mock cells (ctrl) versus CAIX-KO cells either in normoxic conditions or in 1% hypoxia. The cell proliferation was expressed as the cell index, and determined by calculating the slope of the line between time points 5–64 h. Both CAIX knockout cell lines substantially decreased cellular proliferation in hypoxia (Figure 4). Notably, MDA-MB-231-CA9-KO cells reduced proliferation even in normoxia, probably due to the density-dependent induction of CAIX expression in mock cells in normoxic conditions (visible also in Figure 1C).

### 2.6. CAIX Expression Influences LIN28 Together with Stem Cell Markers

LIN28/*let-7* double negative feedback loop was suggested as a reprogramming-like mechanism which resulted in CSCs [28] and LIN28 expression correlates with ALDH1 in ALDH1+ stem cells of breast cancer [29]. Since we revealed that CAIX-knockout MDA-MB-231-CA9-KO and T47D-CA9-KO cells express lower levels of LIN28 protein (Figure 1C), we also tested the expression of CSC-associated markers NANOG and ALDH1. In correlation with decreased LIN28 expression, both CAIX-knockout cell lines exhibited more than 50% reduction of ALDH1 and NANOG mRNAs (Figure 5A). These data indicated that CAIX is relevant for sustaining breast cancer stemness through the regulation of LIN28 expression. Oppositely, overexpression of CAIX in stably transfected MDA-MB-231 cells leads to the enhancement of ALDH1, and NANOG expression, as well as LIN28 (Figure 5B).

## 3. Discussion

Several studies have previously discussed the role of hypoxia-inducible CAIX in cancer, including breast cancer, as a pH regulator, important for adaptation to hypoxia. In the present study, our results show that depletion of CAIX in hypoxic breast cancer cell lines increases several *let-7* miRNAs, with a subsequent decrease of LIN28, glycolytic metabolism and NF-κB activity (see the proposed model in Figure 6).

The *let-7* family members, widely viewed as tumor suppressor microRNAs (targeting multiple oncogenes such as HMGA2, c-Myc, RAS, and cyclin D1) [30], are frequently reduced in cancer, which correlates with increased tumorigenicity and poor prognosis. Mature *let-7* family members have also been reported to be key suppressive regulators of LIN28 expression, by direct binding to the 3′-untranslated region of LIN28 [31]. On the other hand, LIN28A/B can block the biogenesis of *let-7* [12]. Thus, the LIN28/ *let-7* axis is considered a double-negative feedback loop in the regulation of various biological functions.

Nuclear factor-κB (NF-κB), a transcription factor that regulates a battery of genes, critical for immunity, cell proliferation, and tumor development, rapidly reduces *let-7* microRNA levels [11] and enhances expression at the LIN28B promoter [32]. According to our microarray results, transient silencing of CAIX downregulates NF-κB, together with upregulating the *let-7* family members, mentioned above. Decreased NF-κB transactivation was also confirmed by luciferase reporter assay in hypoxic CAIX-knockouts, displaying also decreased levels of LIN28B; this correlates with the determination that CAIX expression is required for the activation of NF-κB in hypoxic breast cancer cells [23].

Tumor cells adapt their metabolism in nutrient-limited conditions by shifting the balance of energy production away from oxidative metabolism to a more glycolytic source, with increased production of lactic acid. Hypoxia-inducible factor-1 (HIF-1), also known as a master regulator of cancer cell metabolism [33], regulates glycolysis under hypoxic conditions, also through the activation of PDK1 [34,35]. The mechanism by which tumor cells have reduced mitochondrial oxidation is presumed to be through the hypoxic reduction of pyruvate dehydrogenase (PDH) activity by HIF-1-induced PDK1, which is essential for inhibitory phosphorylation of PDH E1α at serine 232. Inactivating phosphorylation of PDHA via PDK1 inhibits the conversion of pyruvate to acetyl-CoA. HIF1α-directed PDK1 upregulation is the manner of metabolic adaptation to hypoxia, which leads to the attenuation of mitochondrial function and the TCA cycle along with enhanced glycolysis and lactate production [36]. Our results show that CAIX suppression leads to reduced PDK1 expression and a corresponding decrease in phosphorylation of PDH at Ser-232. Together with reduced production of lactate in breast carcinoma CAIX-knockouts, these data indicate that elimination of CAIX expression downregulates glycolysis in hypoxic breast cancer cell lines and concomitantly attenuates inhibition of oxidative phosphorylation through PDK1/PDH-Ser232 phosphorylation. Thus, CAIX expression could contribute to hypoxic metabolic adaptation through the regulation of PDK1 level via LIN28.

The evolution of hypoxia- and acid-resistant phenotypes within tumor mass is critical for the development of invasive cancer diseases [4]. Thus, proteins responsible for the adaptation to hypoxia and acidosis are promising anti-cancer targets. The role of CAIX in pH regulation and acidification of the tumor microenvironment is based on its enzymatic activity. The underlying mechanism includes CAIX-generated bicarbonate ions that directly feed bicarbonate transporters for the neutralization of intracellular pH [37,38], and simultaneously produced protons support extracellular acidosis, particularly in hypoxic tumors [39]. Inadequate buffering of internal cell pH via knockdown of CAIX might further modify several other signaling pathways. Acidosis induces the production of reactive oxygen species (ROS), activates AKT and NF-κB [40], in the cascade specific to cancer cells. Sensing of acidic pH may enable cells to rapidly reduce mTORC1 activity to temporarily restrain energy-consuming anabolic processes in response to a variety of metabolically stressful conditions [41]. Experiments with attenuation of CAIX activity using CAIX-specific inhibitors identified the mTORC1 axis downstream of CAIX as a critical pathway in the regulation of cancer stem cell function [35]. In addition, the treatment of glioblastoma xenografts by CAIX inhibitor SLC-0111 in combination with temozolomide significantly decreased expression of stem cell markers and reduced the percentage of brain tumor-initiating cells (TICs) and neurosphere formation capacity [42]. There are several papers reporting improved antitumor response of combinatorial therapy targeting CAIX enzymatic activity and conventional chemotherapy, angiogenesis, PD-1 blockade, or mTOR pathway [42,43,44,45,46]. Neutralization of CAIX-mediated tumor acidity enhanced the effectivity of monotherapies and significantly inhibited tumor growth and metastasis. The effect of pH changes on phosphorylation of cellular proteins has also been reported [47]: acidification of culture medium results in a considerable increase in the phosphorylation. Moreover, LIN28 function is also affected by phosphorylation, since Ser-200 phosphorylation increases its protein stability [48]. Here we showed that exposure to acidic media pH 6.7 in our experiments increases the level of LIN28 in hypoxic MCF7 and MDA-MB-231 cells. Concomitant upregulation of CAIX together with LIN28 and PDK1 induced by acidosis indicates the mechanism by which CAIX influences the expression of LIN28 in hypoxia. Considering the hypothesis that evolution of glycolytic and acid-resistant phenotypes are key events in progression from in situ to invasive cancer [4], we suppose that hypoxia-induced CAIX which facilitates survival of cancer cells in harsh hypoxia-acidosis-related TME can contribute to the selection of the Warburg effect phenotype through the regulation of LIN28/*let-7* axis which target PDK1 expression and enhances glycolytic metabolism.

Furthermore, we showed that cells with suppressed CAIX display attenuated cell proliferation. These results are in agreement with the fundamental role of LIN28 in breast cancer cells promoting and sustaining proliferation, with direct potentiation of cellular metabolism [21]. It has been demonstrated that the LIN28/*let-7* axis regulates glycolysis via PDK1 under normoxia, and moreover, that this regulation is independent of HIF-1 [14]. Additionally, attenuated proliferation could be the result of reduced mTOR activity, which was proved in CAIX suppressed/inhibited cancer cells [44,49,50]. As mTOR signaling pathway is supervised by intracellular pH, and phosphorylation of ribosomal protein S6 and protein synthesis is downregulated by intracellular acidosis [41,51]. CAIX could represent an upstream regulator of mTOR signalization and subsequent proliferation.

Increasing evidence suggests that LIN28 may also be a master regulator controlling the pluripotency of embryonic stem cells. LIN28, together with OCT4, SOX2, and NANOG (the “reprogramming factors”), can reprogram somatic cells to induced pluripotent stem cells [22,28]. Multiple studies have reported that LIN28 binds to mRNAs and controls their stability and translation. LIN28 is a marker of cancer stem cells, which contribute to tumor relapse after conventional treatment, including chemotherapy. In addition, a requirement for CAIX expression and activity has been demonstrated for the maintenance of the mesenchymal phenotype, a characteristic feature of breast CSCs [49,52]. Inhibition of CAIX enzymatic activity in orthotopic breast tumors of mice leads to loss of CSCs [49]. Moreover, tumor-initiating cells (TICs) isolated from pancreatic cancer patient positive for CSCs markers EpCAM+/CD44+/CD24+ display high CAIX expression. Silencing of CA9 in these EpCAM+/CAIX (high) population abolished their capability to initiate tumors in PDX models [53].

Interestingly, LIN28 plays a crucial role in the maintenance of ALDH1+ tumor cells, which represent stem-like subpopulation with the cancer initiation competence [29]. Clinical evidence shows that a high percentage of ALDH1 positive cells in many types of epithelial tumors, including breast and pancreatic, is associated with a poorer clinical outcome. Clinical data from 94 breast cancer patients reveal that CD44^+^/CD24^−/low^ tumor cells were more common in CAIX^+^ than in CAIX^−^ tumors. Altogether, these data indicate that CAIX is essential for the maintenance of stem cell phenotype, and we proved the involvement of CAIX in regulating stemness through the modulation of the LIN28/*let-7* signaling axis. Although the precise mechanism has not been elucidated so far, long noncoding RNAs (lncRNAs) associated with hypoxic/acidic microenvironment could represent such mediators between CAIX and LIN28/*let-7*. CAIX-affected microenvironment characterized by acidic tumor milieu with slightly alkaline intracellular pH could regulate the expression of specific long noncoding RNAs or miRNAs which are involved in many biological processes during cancer development. In the study of prostate and breast tumor cells, Riemann et al. [54] identified several miRNAs (including *let-7*) with altered expression during extracellular acidosis. Furthermore, the microenvironment-related expression for several lncRNAs like H19, HOTAIR, NEAT1, linc-ROR, UCA1 or lncRNA-HAL was confirmed, whilst some of them are known to regulate stemness, glycolysis or PDK1 [55,56]. Thus, targeting CAIX-positive cancer cells also hits CSCs subpopulation, which is critical in therapeutic resistance and cancer progression.

Hypoxia-induced CAIX plays an important role in the acidic pH neutralization within the tumor microenvironment. In the present paper, we revealed CAIX-regulation of the LIN28/*let-7* axis in hypoxic breast cancer cells. LIN28/*let-7* may exhibit either oncogenic activities or tumor-suppressive activities in hypoxia and acidic tumor microenvironment, depending on the cellular context, tumor types, or specific microenvironmental stress. The therapeutic modulation of these hypoxia-mediated pathways might be a promising new approach for preventing and/or treating cancer.

## 4. Materials and Methods

### 4.1. Cell Cultures

For the initial microarray study of differential gene expression, we used hypoxic MCF7 (ATCC-HTB-22, delivered by LGC Standards, Teddington, UK) breast cancer cells transiently transfected with control-siRNA or CA9-siRNA. Following the MCF7 microarrays were in the experiments included cell lines MDA-MB-231 (ATCC-HTB-26, LGC Standards) and T47D (ATCC -HTB-133, LGC Standards) to avoid cell-line-specific effects of CAIX suppression. All used cell lines express low or no amount of CAIX protein in normoxia, but its expression is strongly upregulated in hypoxia (with moderate upregulation by high cell density). MDA-MB-231 cells represent basal-like triple-negative breast cancer with the expression of mutant p53; MCF7 and T47D cell lines represent luminal type (with wt-p53, or mutated p53, respectively).

All cell lines were cultured in DMEM with 10% fetal calf serum (Biochrom, Berlin, Germany) at 37 °C in humidified air with 5% CO_2_. Hypoxic experiments were performed in a hypoxic workstation (Ruskinn Technologies) in a 1% O_2_, 2% H_2_, 5% CO_2_, 92% N_2_ atmosphere at 37 °C. Acidic medium with defined pH was prepared using HCO_3_--free DMEM (Sigma Aldrich, Steinheim, Germany) where desired pH value was adjusted by adding an appropriate amount of NaHCO_3_ (4.38 mM NaHCO_3_ in 5% CO_2_ atmosphere for pH 6.7), with standard glucose 4.5 g/L.

Transfections were performed using the TurboFect reagent (ThermoScientific, Waltham, MA, USA) according to the manufacturer’s protocol. Transiently transfected cells were analyzed 24–72 h post-transfection. For transient CAIX expression, we used in-house generated pcDNA3.1-FL-CA9. Stable transfectants were selected in G418 (800–1000 µg/mL).

Inhibition of the enzymatic activity of CAIX were MCF7 cells cultured in 1% hypoxia (1% O_2_, 72 h) with or without 100 µM HSFA. (Note that we used CAIX for the carbonic anhydrase IX protein, and CA9 for corresponding gene relationship, e.g., mRNA, siRNA, plasmids, and particular knockout cell lines.)

### 4.2. Microarray Analysis

Gene expression changes were analyzed by Human Gene 1.0ST Array, Affymetrix, as previously described in Radvak et al. [57]. Microarray analysis was performed in the Laboratory of Genomics and Bioinformatics, Institute of Molecular Genetics of the Academy of Sciences, Czech Republic. The analysis was performed in three replicates. The raw data were evaluated with Partek Genomics Suite (Partek Inc., St. Louis, Missouri, USA).

### 4.3. Transient Silencing

For transient silencing, we used siRNA smart pool system (Dharmacon, Lafayette, CO, USA) consisting of four different siRNA sequences, thereby reducing off-targets. Transfection was performed according to the manufacturer’s recommendations (DharmaFECT siRNA Transfection Protocol, ThermoScientific). Cells were transfected with siCA9 to attenuate CAIX expression and with control nontargeting si-CTRL at 20 nM final siRNA concentration.

### 4.4. Generation of Knockout Cell Line with CRISPR/Cas9 System

The CRISPR/Cas9 system was prepared in our lab [58] with CA9-guide-RNA targeting the PG domain of the CA9 gene. The complementary oligonucleotides for guide RNAs (gRNAs): 5′-CACCGATGCAGGAGGATTCCCCCTT-3′ and 3′-CTACGTCCTCCTAAGGGGGAACAAA-5′ were annealed and cloned into pX459 CRISPR/Cas9-Puro vector (Addgene, Cambridge, MA, USA). MDA-MB-231, T47D, and MCF7 cells were transfected with either pX459/gRNA or pX459 control according to the manufacturer’s instructions. Two days after transfection, cells were treated with 1 μg/mL of puromycin for three days. After two weeks, colonies were isolated and analyzed by Western blot.

### 4.5. Immunoblotting

Cell lysates were prepared in RIPA buffer as described previously [59]. Proteins were quantified by the BCA kit (Pierce, Rockford, IL, USA), separated in 10% SDS-PAGE gels under reducing conditions and transferred onto the PVDF membrane, as described elsewhere [59].

Primary antibodies (listed in Table 2) were detected with HRP secondaries (1:5000, Dako, Glostrup, Denmark) and visualized with ECL or with fluorochrome-labeled secondary antibodies (Li-Cor, Lincoln, NE, USA) visualized on a Li-Cor Odyssey. Signal was quantified using the ImageJ software.

### 4.6. Real-Time Quantitative PCR (qPCR)

Total RNA was isolated using TRI-reagent (Sigma). Reverse transcription of RNA and qPCR were performed as described previously [59], used primers are listed in Table 3. Amplifications were performed in triplicate in 3–5 independent experiments.

Confirming the microarray data-RNA isolation and qPCR of miRNA (*let-7*c, *let-7*d), was performed as previously described in Jurkovicova et al. [61].

### 4.7. Lactate Measurement

Media from tested cells were diluted 1:10 and lactate was enzymatically detected using the Lactate Assay Kit (MAK064, Sigma Aldrich), or diluted 1:50 and detected by Lactate-GloTM Assay (Promega, Madison, WI, USA) according to the manual. In all samples, lactate production was normalized to protein concentration. The experiment was repeated three times.

### 4.8. xCELLigence Real-Time Cell Assay (RTCA)

RTCA was performed as described previously [59]. MDA-MB-231 and T47D mock cells, MDA-MB-321-CA9-KO and T47D-CA9-KO were plated in quadruplicates at 7 × 103 cells/well (adjusted to the final volume of 200 µL). The impedance was recorded in 15 min intervals for 64 h, either in control conditions or in 1% hypoxia. Recorded values were presented as cell index (CI) calculated as a relative change in electrical impedance.

### 4.9. Reporter Assay for NF-κB

MCF7 and MDA-MB-321 cells positive for CAIX, and knockout MCF7-CA9-KO and MDA-MB-321-CA9-KO were transiently transfected with pGL3-empty or pGL3-NF-κB luciferase reporter vector. The day after transfection, cells were trypsinized and plated in triplicates into 24-well plates. Cells were allowed to attach and transferred to 1% hypoxia or maintained in normoxia for an additional 24 h. Reporter activity was measured using the dual-luciferase reporter assay system (Promega, Madison, WI, USA) according to the manufacturer’s instructions, and luciferase activity was normalized against Renilla activity. The experiment was repeated three times (n = 3).

### 4.10. Immunofluorescence

Cells grown on glass coverslips were fixed in 4% paraformaldehyde for 10 min, permeabilized with 0.1% Triton X-100 and further processed as described previously [3]. Samples were analyzed by Zeiss LSM 510 Meta confocal microscope, with objective 40×, at the same microscope settings for all samples.

### 4.11. Statistical Analyzes

Results were analyzed by a two-tailed unpaired t-test (Student’s t-test), and *p* < 0.05 was considered significant. *p* < 0.05 is denoted as *, *p* < 0.01 as ** and *p* < 0.001 as ***. Error bars represent mean ± standard deviation (SD). All experiments were repeated a minimum of three times.

## 5. Conclusions

In summary, we identified that CAIX, a hypoxia-induced pH regulator, regulates the LIN28/*let-7* axis-mediated metabolic shift and stemness in breast cancer cells. Suppression of CAIX by siRNA or CRISPR/Cas9 decreases the level of LIN28 with subsequent downregulation of PDK1 and attenuation of glycolysis. In addition, MDA-MB-231 and T47D CAIX-knockouts exhibit a LIN28-correlated reduction of CSC-associated markers NANOG and ALDH1. Oppositely, overexpression of CAIX in stably transfected MDA-MB-231 cells leads to the enhancement of ALDH1, NANOG, and LIN28. These findings support the view that CAIX is an important component of cancer phenotype participating in key hypoxia and acidosis-associated pathways of cancer progression.

## Figures and Tables

**Figure 1 ijms-21-04299-f001:**
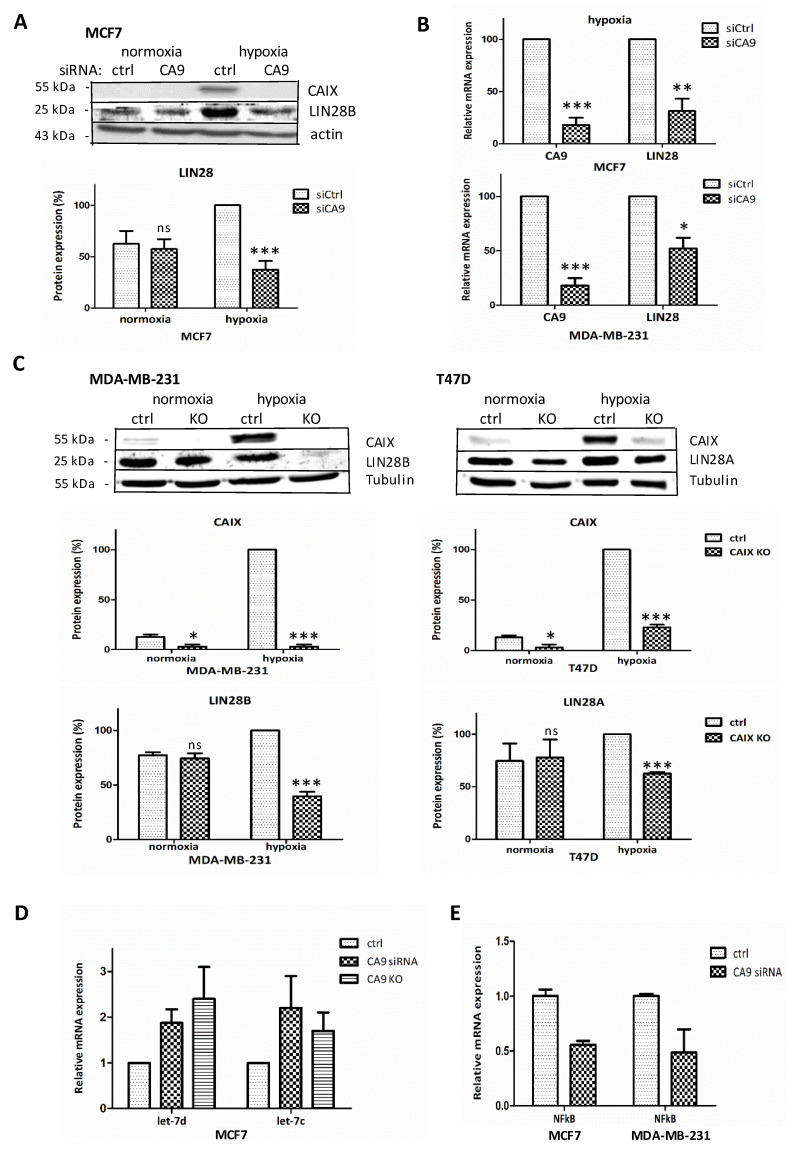
CAIX suppression affects LIN28 expression. (**A**) Representative Western blot of normoxic and hypoxic MCF7 cells transiently transfected with siRNA-control/siRNA-CA9, detecting CAIX, actin, and LIN28. Signal intensities from three different immunoblots were quantified using the ImageJ software, normalized to actin, and presented as an average percentage of hypoxic control-siRNA (=100%). (**B**) Quantitative PCR analysis of CA9 and LIN28 mRNA expression in hypoxic *CA9*-silenced cells normalized to actin, presented as a percentage of the expression in cells transfected with control-siRNA (siCtrl = 100%). The results represent the mean of three independent biological experiments done in triplicates. (**C**) Representative Western blot of CAIX and LIN28 (homolog B for MDA-MB-231, homolog A for T47D) in MDA-MB-321-CA9-KO and T47D-CA9-KO cell lines. Average signal from 3 different Western blots quantified by the ImageJ software, normalized to tubulin, and presented as a percentage of hypoxic control (=100%). (**D**) The effect of CAIX-depletion on *let-7* expression (qPCR) in MCF7, confirming microarray data presented in Table 1, extended with confirmation in KO approach. (**E**) NF-κB expression (qPCR) in si-CA9 suppressed MCF7 (according to microarray data presented in Table 1) and also in MDA-MB-231 cells. *p* > 0.05 was considered nonsignificant (ns), *p* < 0.05 is denoted as *, *p* < 0.01 as ** and *p* < 0.001 as ***.

**Figure 2 ijms-21-04299-f002:**
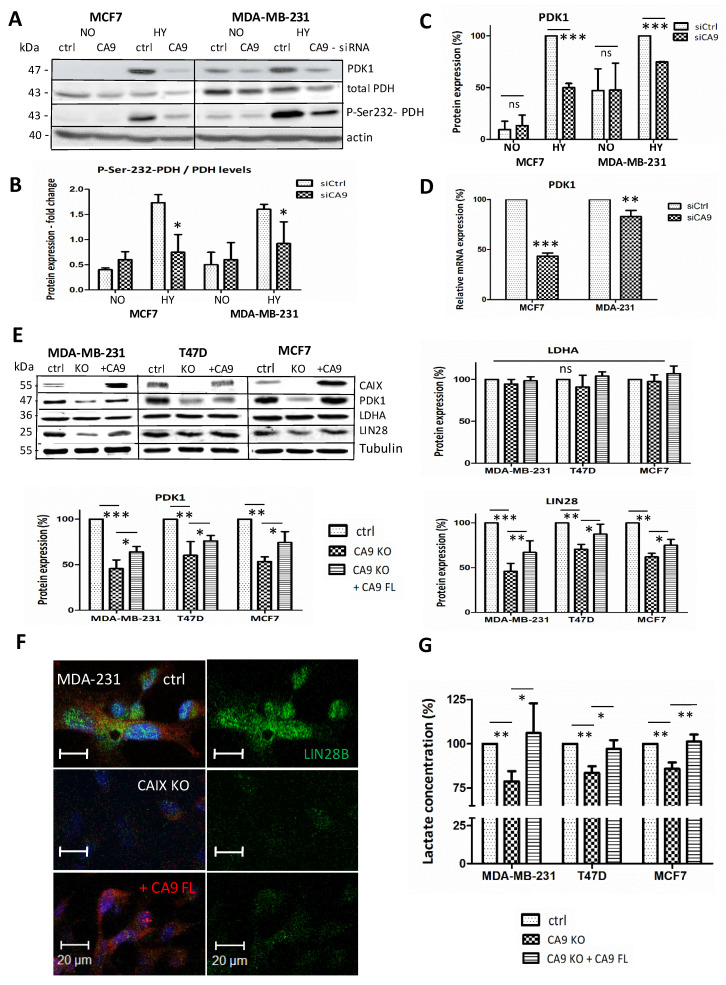
The effect of CAIX suppression on the glycolytic pathway. (**A**) Representative Western blot of PDK1 (Pyruvate dehydrogenase kinase 1), total PDH (Pyruvate dehydrogenase), and pSer232-PDH in silenced MCF7 and MDA-MB-231 cells, incubated in normoxia (NO) or hypoxia (HY), with graphic illustration (**B**) of the proportion of Ser-232- phosphorylated-PDH to total-PDH (signal quantified via ImageJ software). (**C**) PDK1 signal from three different immunoblots quantified via ImageJ software and normalized to internal control. Hypoxic control was set as 100%. (**D**) Quantitative PCR of PDK1 mRNA in hypoxic control and CA9-silenced cells normalized to actin, presented as a percentage of control expression. The results represent the mean from three independent biological experiments done in triplicates. (**E**) Representative Western blots and their signal quantification of hypoxic control cells, CAIX-knockouts, and knockouts with transiently restored CAIX expression, detecting PDK1, LDHA, and LIN28 (the bottom part of the membranes were divided for anti-LIN28 homolog B for MDA-MB-231 and MCF7, homolog A for T47D). Signals were quantified from five different immunoblots via ImageJ software, and normalized to tubulin. (**F**) Immunofluorescence of hypoxic MDA-MB-231, their CAIX-knockout, and CA9 transfected KO cells, with CAIX (red) and LIN28B (green) staining, (Zeiss LSM 510 Meta confocal microscope with objective 40×). (**G**) Lactate concentrations in cell culture media from CAIX-knockout MDA-MB-231, T47D, and MCF7 cells, and knockout cells transiently transfected with CA9-FL, determined by two types of enzymatic lactate assay. The concentration of lactic acid in the samples was calculated from the average of three different experiments (done in triplicate measurements) and presented as a percentage of control expression (=100%). *p* > 0.05 was considered nonsignificant (ns), *p* < 0.05 is denoted as *, *p* < 0.01 as ** and *p* < 0.001 as ***.

**Figure 3 ijms-21-04299-f003:**
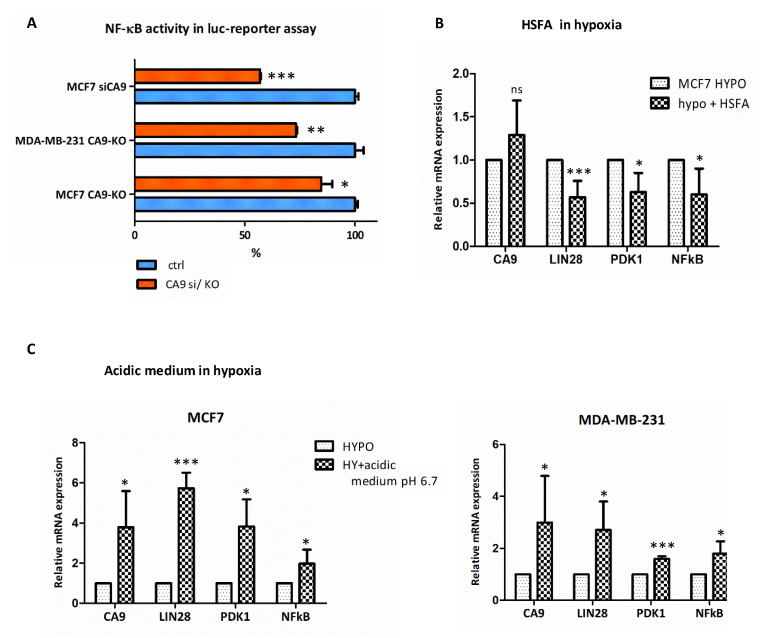
Relationship of CAIX to NF-κB. (**A**) CAIX depletion reduces NF-κB activity. Luciferase assay monitoring the activity of the NF-κB reporter vector in hypoxic cells with induced CAIX expression (100%) in comparison to CAIX-knockout MCF7-CA9-KO, MDA-MB-321-CA9-KO, and transiently silenced MCF7-siCA9 cells. (**B**) Impact of CAIX enzymatic inhibition by HSFA (100 µM) on CA9, LIN28B, PDK1, and NF-κB mRNA expression in hypoxic MCF7 cells. (**C**) Effect of modified acidic medium pH 6.7 on the set of the same genes mRNA expression in hypoxic MDA-MB-231 and MCF7 cells. mRNA expression was normalized to actin and presented as a percentage of expression in the control medium. The results represent the mean of three independent biological experiments performed in triplicates. *p* > 0.05 was considered nonsignificant (ns), *p* < 0.05 is denoted as *, *p* < 0.01 as ** and *p* < 0.001 as ***.

**Figure 4 ijms-21-04299-f004:**
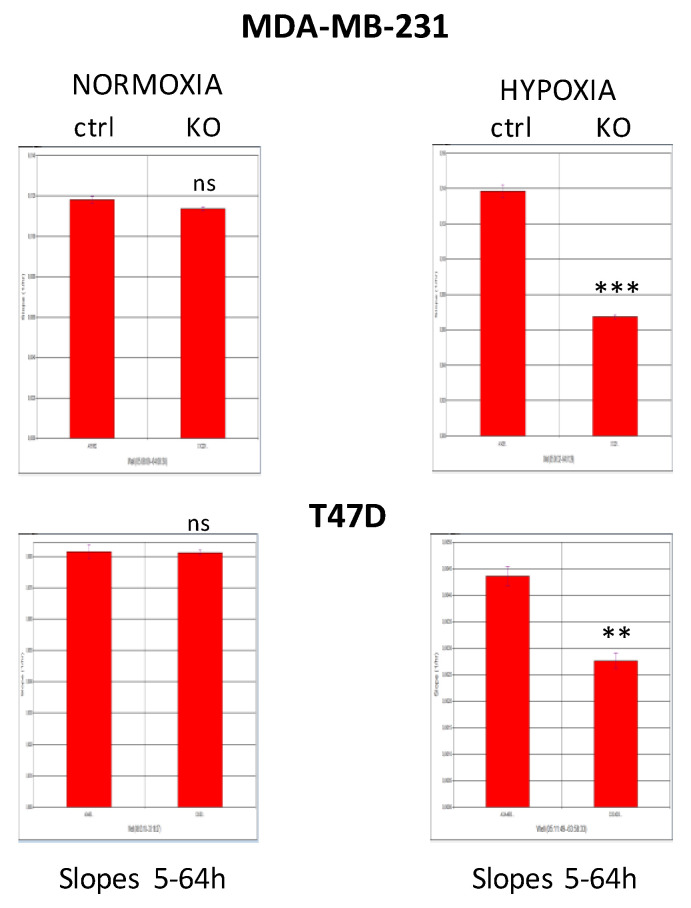
The effect of CAIX knockdown on cell proliferation. MDA-MB-231 and T47D mock cells (ctrl) and their CAIX KO counterparts were cultured in normoxic or hypoxic (1% O2) conditions. Proliferation was monitored in real-time using the xCELLigence system and expressed as the cell index. Slopes derived from the measurement data indicate the rate of cell growth between time points 5–64 h. The experiment was repeated two times in quadruplicates. *p* > 0.05 was considered nonsignificant (ns), *p* < 0.01 is denoted as ** and *p* < 0.001 as ***.

**Figure 5 ijms-21-04299-f005:**
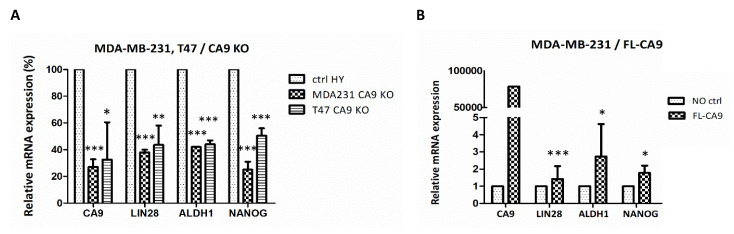
(**A**) The effect of CAIX knockout on mRNA expression of LIN28 and some stem cell markers in MDA-MB-231-CA9-KO and T47D-CA9-KO cells. (**B**) Impact of CAIX overexpression in stably transfected MDA-MB-231 on the expression of the same genes set. mRNA expression was normalized to actin and was presented as a percentage of hypoxic control parental cells. The results represent the mean from three independent biological experiments performed in triplicates. *p* > 0.05 was considered nonsignificant (ns), *p* < 0.05 is denoted as *, *p* < 0.01 as ** and *p* < 0.001 as ***.

**Figure 6 ijms-21-04299-f006:**
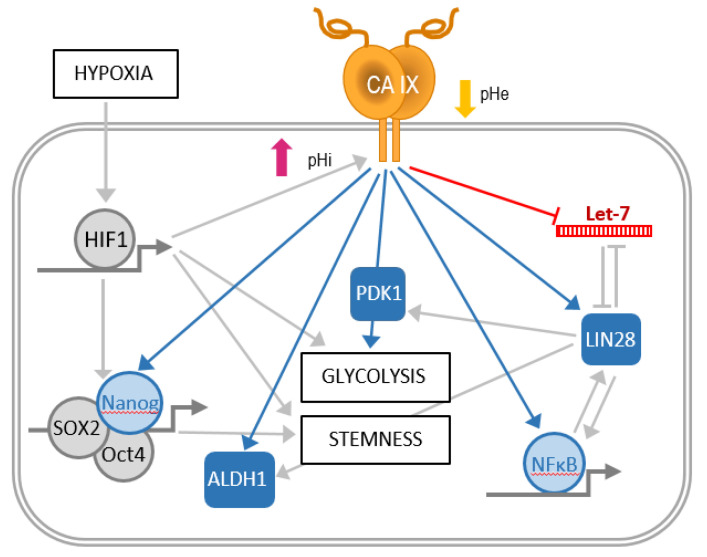
Schematic illustration of the proposed CAIX contribution to metabolic adaptation and stemness of cancer cells. Manipulation of CAIX levels via RNA interference, gene editing, or ectopic expression in hypoxic breast cancer cells supports the role of CAIX in the downregulation of *let-7* microRNA family members, resulting in upregulation of LIN28 RNA-binding protein and activation of NFκB transcription factor. The expression of CAIX is also associated with increased level and activation of PDK1 kinase, which is important for the glycolytic metabolism. Moreover, CAIX contributes to higher levels of SOX2, Oct4, and Nanog transcription factors and ALDH1 protein participating in cancer cell stemness. Interestingly, CAIX, HIF-1, and LIN28 pathways are interconnected through multiple relationships and feedback loops, suggesting their complex regulatory network, in which CAIX can presumably play a role as a mediator of responses to hypoxia and acidosis. Color coding: Molecules in blue color were identified as positively regulated by CAIX, the one in red color is negatively regulated by CAIX. The arrows indicate upregulation, while the T line indicates downregulation. Gray arrows represent relationships already known from the literature, while blue arrows and the red T line represents newly identified relationships described in this study.

**Table 1 ijms-21-04299-t001:** A subset of differentially expressed genes from microarray analysis, relevant to this study.

Symbol	Gene Name	logFC	*p*-Value	fdr
MIRLET7D	microRNA *let-7d*	0.51	0.0015	0.34
MIRLET7C	microRNA *let-7c*	0.5	0.017	0.36
MIR125B2	microRNA *125b-2*	0.43	0.04	0.4
MIRLET7F1	microRNA *let-7f-1*	0.43	0.045	0.4
NFKB1	*nuclear factor kappa B subunit 1*	–0.83	0.039	0.4

**Table 2 ijms-21-04299-t002:** List of primary antibodies.

Antigen	Host	Dilution	Company
actin	M	1:1000	Cell Signaling 8H10D10
tubulin	Rb	1:2000	Abcam 4074
CAIX	M	1:4	in-house M75 [60]
total PDH E1α	M	1:6000	MitoSciences
pSer232-E1α	Rb	1:3000	EMD Chemicals
PDK1	Rb	1:1000	ENZO
LIN28A	Rb	1:1000	Cell Signaling A177
LIN28B	Rb	1:500	Abcam 71415

**Table 3 ijms-21-04299-t003:** List of primers (5′–3′).

B-Actin S	CCAACCGCGAGAAGATGACC	B-actin A	GATCTTCATGAGGTAGTCAGT
CA9 S	CCGAGCGACGCAGCCTTTGA	CA9 A	GGCTCCAGTCTCGGCTACCT
LIN28A F	GAGTGAGAGGCGGCCAAAA	LIN28A R	TGATGATCTAGACCTCCACAGTTGTAG
LIN28B F	TGATAAACCGAGAGGGAAGC	LIN28B R	TGTGAATTCCACTGGTTCTCC
PDK1 S	ATTGGAAGCATAAATCCAAACTG	PDK1 A	CGGTCACTCATCTTCACAGTC
LDHA F	TGGCAGCCTTTTCCTTAGAA	LDHA R	ACTTGCAGTTCGGGCTGTAT
ALDH1 F	CGGGAAAAGCAATCTGAAGAGGG	ALDH1 R	GATGCGGCTATACAACACTGGC
NANOG S	GCAAATGTCTTCTGCTGAGATGC	NANOG A	AGCTGGGTGGAAGAGAACACAG
NF-κB/p65 F	GACCTGAATGCTGTGCGGC	NF-κB/p65 R	ATCTTGAGCTCGGCAGTGTT

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
