# Peer review of "CAIX-Mediated Control of LIN28/let-7 Axis Contributes to Metabolic Adaptation of Breast Cancer Cells to Hypoxia"

_ijms, 2020, doi:10.3390/ijms21124299_

Round 1
Reviewer 1 Report
COMMENTS:
Results Fig.1:
- The authors suggested that CAIX was upregulated under hypoxia and that LIN28B was under CAIX control. Looking at W.B. analysis, I don’t see CAIX expression in MCF-7 cell line under normoxia and very low levels in the other two cell lines. So, why they applied gene silencing or Crisp/Cas9 gene know-out in normoxia condition is not clear.
- Why Crisp/Cas9 reduced CAIX expression and not total inhibition of its expression, as shown in Fig.1C.
- Why did they use two different housekeeping genes and proteins for PCR and W.B normalization (b-actin for MCF-7 and Tubulin for MDA and T47D cell lines)?
- The quality of W.B. should be improved (as in LIN28B bands in Fig.1A).
- No statistical analysis was represented in siCtrl densitometric analysis
Results fig.2:
- Which is the difference between total PDH and PDH in Fig.2 A?
- The CAIX protein expression in Ctrl cells under hypoxia is lower than +CA9-transfected cells, however, the PDK1 protein is higher in Ctrl compared with +CA9-transfected cells. If CAIX induces PDK1 expression, why its expression is lower in +CA9?
- As for Fig1, why different normalization proteins (actin and tubulin)?
Results Fig3:
- NfkB activity was reduced to 84.7% and 73% in CA-9 KO MCF-7 and MDA cell lines, and the NfkB activity in siRNA cells is lower than KO-cells. So the question is: gene silencing work better than gene knock-out?
The Cell lines were transiently o stable transfected with CA-vector? In 3.6 section is stable and in 3.2 is transient.
Author Response
Response to Reviewer 1 Comments
We want to thank the reviewer for his/her valuable comments. Please, see our responses below. The changes made in the text are marked in yellow and clearly described in the answers given below.
Reviewer 1
Results Fig.1:
Point 1: The authors suggested that CAIX was upregulated under hypoxia and that LIN28B was under CAIX control. Looking at W.B. analysis, I don’t see CAIX expression in MCF-7 cell line under normoxia and very low levels in the other two cell lines. So, why they applied gene silencing or Crisp/Cas9 gene know-out in normoxia condition is not clear.
Response 1: All used cell lines express low or no amount of CAIX protein in normoxia, but its expression is strongly upregulated in hypoxia (with moderate upregulation by high cell density). (inserted lines 83-84). Actually, normoxic samples with CA9 silencing serve as a negative control to confirm that effects mediated by siRNA targeting CA9 (in hypoxia) are really consequences of CAIX downregulation and are not the result of off-target effects.
Point 2: Why Crisp/Cas9 reduced CAIX expression and not total inhibition of its expression, as shown in Fig.1C.
Response 2: The efficiency of CRISPR/Cas9 as well as siRNA systems to alter gene expression is not 100% and vary in a wide range as it is reviewed in several papers e.g. in Unniyampurath et al., IJMS 2016; Lino et al., Drug Delivery, 2018; Duda et al., Nucleic Acids Res. 2014; Liang et al., J. Biotechnol. 2015; Swiech et al., Nat. Biotechnol. 2015.
Point 3: Why did they use two different housekeeping genes and proteins for PCR and W.B normalization (b-actin for MCF-7 and Tubulin for MDA and T47D cell lines)?
Response 3: To detect proteins in WB we used 2 different systems namely enhanced chemiluminescence (HRP-conjugated secondary antibodies) or the LI-COR Odyssey system (fluorescently-conjugated secondary antibodies). β-actin was used with ECL, and tubulin was used with LI-COR system.
Point 4: The quality of W.B. should be improved (as in LIN28B bands in Fig.1A).
Response 4: Thank you for your comment. Different approach to protein visualization is the reason behind different quality of W.B.: 1.A was visualized with ECL whilst 1.C with LI-COR Odyssey system, which gives a different background.
Point 5: No statistical analysis was represented in siCtrl densitometric analysis
Response 5: Densitometric analysis 1.B was presented as percentage of the expression in cells transfected with control-siRNA (siCtrl=100%), in 1.C it was presented as percentage of hypoxic control (=100%).
Point 6: Which is the difference between total PDH and PDH in Fig.2 A?
Response 6: We apologize for this mistake (designation ”P-Ser232-PDH” was splitted to two lines during line spacing). We corrected designation in Figure 2A.
Point 7: The CAIX protein expression in Ctrl cells under hypoxia is lower than +CA9-transfected cells, however, the PDK1 protein is higher in Ctrl compared with +CA9-transfected cells. If CAIX induces PDK1 expression, why its expression is lower in +CA9?
Response 7: The level of CAIX expression in transiently transfected cells does not perfectly reflect the situation of natural hypoxic induction of CAIX. As we still do not know the precise mechanism by which CAIX affects let7/LIN-28 level which is partially responsible for PDK1 expression, and we do not think that CAIX works as a transcription factor, we do not expect that the PDK1 level will increase proportionally to CAIX level.
Point 8: As for Fig1, why different normalization proteins (actin and tubulin)?
Response 8: Please, see the response 3.
Point 9: Results Fig3: NfkB activity was reduced to 84.7% and 73% in CA-9 KO MCF-7 and MDA cell lines, and the NfkB activity in siRNA cells is lower than KO-cells. So the question is: gene silencing work better than gene knock-out?
Response 9: Not necessarily, CAIX-knockouts MCF7-CA9-KO, MDA-MB-321-CA9-KO, may differ from parent cells in transfection efficiency with reporter NF-κB luciferase vector. Moreover, in case of transiently silenced MCF7-siCA9 cells NFκB activity reflects double transfection (siCA9RNA + reporter NF-κB luciferase plasmid).
Point 10: The Cell lines were transiently o stable transfected with CA-vector? In 3.6 section is stable and in 3.2 is transient.
Response 10: For the evaluation of metabolic consequences of CAIX overexpression we used transient transfection whilst for the examination of stem cell markers expression which reflects long-term changes we used stably transfected cells.
Reviewer 2 Report
The authors mentioned that carbonic anhydrase IX (CAIX) regulates Let-7 miRNAs member, and its downregulation therefore causes the decreased protein and mRNA level of LIN28 in hypoxia. They claim that this regulation results in the adaptation of breast cancer cells to hypoxia. Adaptation of cancer cells to their microenvironment is still very interesting topic to investigate. Therefore, I am convinced that this study is appropriate the publish in the International Journal of Molecular Sciences after major/minor changes.
Major Changes:
- In result 1 (Suppression of CAIX affects the let-7/LIN28 axis): they showed the protein expression level of CAIX and LIN28B in both normoxia and hypoxia, whereas they showed the mRNA expression level of CAIX and LIN2B in only hypoxia. Therefore, I would like to see the mRNA expression level in both normoxia and hypoxia. In this case, I kindly suggest to authors to perform additional experiments considering the journal’s rules.
- In Figure 1: the mRNA and protein expression level of LIN28B and LIN28A was shown in MDA-MB-231 and T47D cell lines. However, for MCF-7 cell lines, only LIN28B was shown.
- Please provide the data about the expression level of both protein and mRNA levels of LIN28A as well.
- Both siCAIX and knockout CAIX groups show not significantly affect on LIN28B protein expression level in normoxia, while it is downregulated dramatically in hypoxia. Why this effect occurs in only hypoxia instead of in normoxia and hypoxia, please discuss this phenomenon in discussion and add new citations about this issue.
- In Figure 1D: For Let-7d and Let-7c; The authors performed the experiments by using both siRNA and knockout CAIX in MCF-7 cell lines. However, for NF-κB; they only performed the experiments by using siRNA in both MCF-7 and MDA-MB-231 cell lines. How they distinguished the experimental design in which needs to be used siRNA and which needs to be used knockout cell lines?
- Why they did not perform knockout experiments in MCF-7 cells in Figure1A and B as they performed in figure 1D. Please arrange your data and experiments with totally the same condition.
Not: Internal control should be at the end of whole proteins. In figure 1 and somewhere else, it was shown between two proteins. Please change the location of the internal control.
Minor Changes:
- The authors wrote CAIX and CA9 as the abbreviation of carbonic anhydrase. Which one is correct? Or is there any difference meaning between them? Please make sure and change it as one of them.
- In figure 1: please mark clearly ‘’normoxia’’ and ‘’hypoxia’’ above of quantitative data. Also, protein name needs to be written in Y-axis of quantitative data
- Line 47-49
The hypothesis mentioned in line47-49 is not appropriate the write in this section of the introduction. It would be better to write in the last part of the introduction. In addition, the hypothesis in this sentence does not show the whole picture that the readers need to understand the main story from this paper. It should be more specific explanations, which will be easy to get the whole picture.
- Line 65-67
Please give some information and new citations about the Warburg effect.
- The discussion part gives us useful information. However, I prefer to read more comments and encountered problems and solutions for the data represented in this paper. Because the data presented in this paper may not convince the readers unless a clear explanation made by the authors, I believe that the discussion part needs to be rewritten according to the new data that they will perform to get published in this journal.
Author Response
Response to Reviewer 2 Comments
We want to thank the reviewer for his/her valuable comments. Please, see our responses below. The changes made in the text are marked in yellow and clearly described in the answers given below.
Reviewer 2
The authors mentioned that carbonic anhydrase IX (CAIX) regulates Let-7 miRNAs member, and its downregulation therefore causes the decreased protein and mRNA level of LIN28 in hypoxia. They claim that this regulation results in the adaptation of breast cancer cells to hypoxia. Adaptation of cancer cells to their microenvironment is still very interesting topic to investigate. Therefore, I am convinced that this study is appropriate the publish in the International Journal of Molecular Sciences after major/minor changes.
Major Changes:
Point 1: In result 1 (Suppression of CAIX affects the let-7/LIN28 axis): they showed the protein expression level of CAIX and LIN28B in both normoxia and hypoxia, whereas they showed the mRNA expression level of CAIX and LIN2B in only hypoxia. Therefore, I would like to see the mRNA expression level in both normoxia and hypoxia. In this case, I kindly suggest to authors to perform additional experiments considering the journal’s rules.
Response 1: All used cell lines express low or no amount of CAIX protein in normoxia, but its expression is strongly upregulated in hypoxia (with moderate upregulation by high cell density). (included: 83-84). Actually, normoxic samples with CA9 silencing serve as a negative control to confirm that effects mediated by siRNA targeting CA9 (in hypoxia) are really consequences of CAIX downregulation and are not the result of off-target effects.
Point 2: In Figure 1: the mRNA and protein expression level of LIN28B and LIN28A was shown in MDA-MB-231 and T47D cell lines. However, for MCF-7 cell lines, only LIN28B was shown.
Response 2: According to the literature data, T47D cell line expresses LIN28A whilst MCF7 and MDA-MB-231 express no or very low level of LIN28A. LIN28B is expressed in MCF7 and MDA-MB-231 (ref. 29 Thornton et al; Lv et al., PLoS ONE, 2012; Wang et al., PLoS ONE, 2013). In our experiments we confirmed these data and therefore, we detected the particular LIN28A or B homologs in used cell lines. We included this explanation into the main text, lines 111-114. In Figure 1C we divided the designation of WB for LIN28 homologs for individual cell lines, as it is also mentioned in figure legends (lines 127), and for Figure 2 in line 171.
Point 3: Please provide the data about the expression level of both protein and mRNA levels of LIN28A as well.
Response 3: Please, see the Response 2.
Point 4: Both siCAIX and knockout CAIX groups show not significantly affect on LIN28B protein expression level in normoxia, while it is downregulated dramatically in hypoxia. Why this effect occurs in only hypoxia instead of in normoxia and hypoxia, please discuss this phenomenon in discussion and add new citations about this issue.
Response 4: CAIX is strongly hypoxia induced protein and it is a direct HIF1α target gene with minimal/no expression under normoxic conditions, thus the effect of CA9-silencing on LIN28 level in normoxia is not expected and would be considered as off-target.
Point 5: In Figure 1D: For Let-7d and Let-7c; The authors performed the experiments by using both siRNA and knockout CAIX in MCF-7 cell lines. However, for NF-κB; they only performed the experiments by using siRNA in both MCF-7 and MDA-MB-231 cell lines. How they distinguished the experimental design in which needs to be used siRNA and which needs to be used knockout cell lines?
Response 5: In our experiments we wanted to show that effects caused by CAIX downregulation are not cell line- and suppression technique-specific. Thus, we used three different breast cancer cell lines (MCF7, MDA-MB-231, T47D) and two different techniques (siRNA, Crisp/Cas9) to downregulate CAIX expression, and we did not perform all experiments with all cell lines and both systems.
To improve the clarity of the result in Figure 1D we divided validation of the effect of CAIX downregulation on let-7 and NF-κB into separate figures (1D for let-7 members, and 1E for NF-κB).
Point 6: Why they did not perform knockout experiments in MCF-7 cells in Figure1A and B as they performed in figure 1D. Please arrange your data and experiments with totally the same condition.
Response 6: Thank you for your comment. Please see the response 5. Results with all three used CAIX-KO cell lines showing the impact on LIN28 level are in Figure 2E where, in addition to the effect of CAIX downregulation, we also show the consequences of restored (transiently transfected) CAIX expression in CAIX-knockouts.
Not: Internal control should be at the end of whole proteins. In figure 1 and somewhere else, it was shown between two proteins. Please change the location of the internal control.
Thank you for your comment. The order of the bands was given by molecular weight but we corrected it according to your recommendation.
Minor Changes:
- The authors wrote CAIX and CA9 as the abbreviation of carbonic anhydrase. Which one is correct? Or is there any difference meaning between them? Please make sure and change it as one of them.
- Response: The explanation is mentioned in the text, please see lines 412-413, and also in Abbreviations line 497.
- In figure 1: please mark clearly ‘’normoxia’’ and ‘’hypoxia’’ above of quantitative data. Also, protein name needs to be written in Y-axis of quantitative data
- Response: Thank you for your suggestion. With respect to the fact that Figure 1 is composed from several graphs, we preferred the presentation of each graph in such a design with the title at the top and description “normoxia’’ and ‘’hypoxia’’ at the bottom.
- Line 47-49
The hypothesis mentioned in line47-49 is not appropriate the write in this section of the introduction. It would be better to write in the last part of the introduction. In addition, the hypothesis in this sentence does not show the whole picture that the readers need to understand the main story from this paper. It should be more specific explanations, which will be easy to get the whole picture.
- Response: Thank you for your comment. The whole picture is summarized in part Conclusions. The hypothesis originally mentioned in lines 47-49 is extended in the Discussion, lines 318-323.
- Line 65-67
Please give some information and new citations about the Warburg effect.
- Response: Additional information and new citations about the Warburg effect are included in lines 70-75.
- The discussion part gives us useful information. However, I prefer to read more comments and encountered problems and solutions for the data represented in this paper. Because the data presented in this paper may not convince the readers unless a clear explanation made by the authors, I believe that the discussion part needs to be rewritten according to the new data that they will perform to get published in this journal.
- Response: We have expanded sections of the Introduction and Results providing more detailed descriptions of experimental methods and results according to your suggestions therefore we believe that discussion will be more understandable for readers.
Round 2
Reviewer 1 Report
no other requests
Reviewer 2 Report
Most of the inquiries had been responded. However, the minor spelling check of English writing still needs to attention.